# Prevalence of Musculoskeletal Disorders and Their Associated Risk Factors among Furniture Manufacturing Workers in Guangdong, China: A Cross-Sectional Study

**DOI:** 10.3390/ijerph192114435

**Published:** 2022-11-04

**Authors:** Yan Yang, Jiancheng Zeng, Yimin Liu, Zhongxu Wang, Ning Jia, Zhi Wang

**Affiliations:** 1Key Laboratory of Occupational Environment and Health, Guangzhou Twelfth People’s Hospital, Guangzhou 510620, China; 2Department of Public Health and Preventive Medicine, School of Medicine, Jinan University, Guangzhou 510632, China; 3National Institute of Occupational Health and Poison Control, Chinese Center for Disease Control and Prevention, Beijing 100050, China

**Keywords:** musculoskeletal disorders, furniture manufacturing workers, risk factors

## Abstract

Objective: This study aims to investigate the one-year prevalence and the associated factors of work-related musculoskeletal disorders (WMSDs) among furniture manufacturing workers in Guangdong, China. Methods: A cross-sectional study of 4181 (2953 males and 1228 females) furniture manufacturing workers was conducted between September 2019 and December 2019. All information about WMSDs was collected by the electronic version of Chinese Musculoskeletal Questionnaires (CMQ). Descriptive statistics and a binary logistic regression model were used to interpret the data. Result: The overall prevalence of WMSDs was 31.57%. The WMSD symptoms most commonly occurred in the neck (16.77%), followed by the shoulders (14.90%), ankles/feet (14.64%), hands/wrists (13.30%), upper back (11.48%), and lower back (10.95%). Multiple logistic regression analysis revealed that several individual, labor organization, and ergonomics-related factors conferred significant risks to WMSDs at different body sites. Conclusions: WMSDs remain the major occupational health problem for furniture manufacturing workers. Hence, some effective and feasible protective measures for furniture manufacturing workers are required in order to alleviate the health burden caused by WMSDs.

## 1. Introduction

Musculoskeletal disorders (MSDs) are injuries affecting the muscles, tendons, peripheral nerves, and vascular system due to a wide range of inflammatory and degenerative changes [1,2]. These painful disorders and symptoms related to the musculoskeletal apparatus are caused by the movements in work activities and are thus named as work-related musculoskeletal disorders (WMSDs) [3]. WMSDs are recognized as one of the most common occupational hazards among industrialized workers, resulting in absence from work, poor working performance, disability, and a decline in the quality of life [4,5,6]. According to data from the Global Burden of Diseases, Injuries, and Risk Factors Study (GBD) released in 2019, musculoskeletal disorders are among the top 20 leading causes of disease burden for all ages, especially the low back pain, ranking 9th among the 10 most important drivers of the increasing burden (i.e., the causes that had the largest absolute increases in the number of DALY (disability adjusted life years) between 1990 and 2019) [7]. There were about 1.71 billion (95% UI: 1.63–1.80) prevalent cases and 149 million (95% UI: 108–199) YLDs (years of life lived with disability) due to MSDs in 2019, globally [8]. WMSDs contribute a significant proportion of cases or YLDs, demonstrating that WMSDs not only result in immense suffering to the physical and mental health of working professionals but also impose a heavy burden on the health care services and, consequently, to our societies [8,9].

The issue of manufacturing worker absenteeism due to WMSDs is also quite prominent. According to the reports of the U.S. Bureau of Labor Statistics [10], production manufacturing workers who sustained WMSDs required a median of 12 days to restore before returning to work, and the resulting lost working hours can cause huge financial losses to employers. Data from German economic sectors indicated that the manufacturing sector suffers the highest economic losses owning to WMSDs, with a EUR 6.45 million loss of production and EUR 10.63 million loss of gross value added [11]. Obviously, WMSDs have brought huge disease and economic burdens to the furniture manufacturing industry.

Although advanced machinery and equipment are now available, the furniture manufacturing industry, as a labor-intensive sector, still requires a large amount of human labor. Several studies of furniture manufacturing workers have reported that WMSDs caused considerable distress to this occupation group, primarily involving the shoulders, hand/wrists, and lower back, with the prevalence ranging from 22.7% to 53.9% [12,13,14,15]. The job of furniture manufacturing workers often involves manual material handling, repetitive movements, force exertion, awkward postures, pinch grips, etc., which have been reported to be associated with the development of WMSDs [16,17]. Therefore, it is reasonable to assume that the development of WMSDs in furniture manufacturing workers is associated with individual, labor organization, and ergonomics-related factors.

At present, the research results from the previous studies [12,13,14,15] may be limited due to the small sample size (N = 100–500), which may not fully depict the actual picture of WMSDs in the entire furniture manufacturing industry. In addition, only a few previous surveys [18,19,20] have been conducted in China, one of the world centers for the furniture manufacturing industry. Several important parameters for WMSDs among Chinese workers (e.g., the most affected sites and the epidemiological factors related to the development of WMSDs) remain largely unknown. Hence, large-scale investigations of the Chinese furniture manufacturing industry are highly in demand in order to provide more solid and accurate data for the prevention and treatment of WMSDs. The purpose of this study was to estimate the prevalence and risk factors related to nine body sites/regions (neck, shoulders, upper back, elbows, wrists/hands, lower back, hips/thighs, knees, and ankles/feet) among furniture manufacturing workers affected with WMSDs in Guangdong, China. 

## 2. Methods

### 2.1. Study Population

This survey was conducted in the Guangdong province of China from September 2019 to December 2019, and the subjects were the furniture manufacturing workers from eleven manufacturing factories of a large company in this province. Guangdong is one of the most important industrial regions in China and has become the largest furniture production zone in China, characterized by a huge production capacity, multi-industry clusters, and a complete industrial supply chain. It is also the largest furniture production zone and exchange center in the Asia-Pacific region. 

### 2.2. Sampling Procedure and Sample Size Calculation

This study adopted the method of cluster random sampling, i.e., randomly selecting a large furniture manufacturing enterprise in Guangdong province and conducting a questionnaire survey involving all on-the-job workers in eleven factories under this enterprise. This manufacturer was a typical one, representing the furniture manufacturers in southern China well. It has a complete industrial chain and a good production capacity. In addition, the workers in this company were well trained and had a good willingness to participate in this study, which alleviated our concern regarding a poor responding rate in this survey. A public web server (https://www.calculator.net/sample-size-calculator.html, accessed on 17 October 2022) was used to perform the sample size calculation. A prevalence of WMSDs of 32.27% among furniture manufacturing workers, estimated by a previous study conducted in China [20], was assumed. With the confidence level of 95% (the type I error of 5%) and a margin of error of 2%, the required sample size was estimated to be 2100. Finally, a total number of 4471 questionnaires were collected, and a total of 4181 valid questionnaires remained after invalid questionnaires were excluded (93.51% efficiency rate).

### 2.3. Inclusion and Exclusion Criteria

Subjects were included in this survey if they met the following criteria: (1) age > 18 years; (2) having at least 1 year of continuous work experience in the furniture manufacturing industry. Pregnant women and people with congenital spinal deformity and/or other musculoskeletal diseases caused by non-work related factors such as trauma, infectious diseases, malignant tumors, etc. were excluded. 

### 2.4. Data Collection

The electronic version of the Chinese Musculoskeletal Questionnaires was used to collect data from the study participants. This electronic questionnaire version was based on the standard Nordic Musculoskeletal Questionnaires (NMQ) developed by Kuorinka et al. [17] in 1987. After some modifications to adapt Chinese characteristics, this self-reported questionnaire has been proven to have good reliability and validity and is widely used to assess musculoskeletal symptoms for Chinese occupation groups [21,22].

The questionnaire consists of three parts. The first part is designed to obtain individual demographic information, including age, gender, weight, height, body mass index (BMI), dominant hand, type of work, career length, education level, marital status, monthly income, physical exercise, smoking habits, physical health status, etc. The second part consists of questions related to information on WMSDs in nine body regions/sites (neck, shoulders, upper back, lower back, elbows, wrists/hands, hips/thighs, knees, and ankles/feet) in the previous 12 months. For each region/site, participants were required to answer the following questions: (1) whether they had a work-related ache, pain, discomfort, or other complaint in the previous 12 months; (2) frequency of pain and/or discomfort; (3) cumulative total time for symptom onset throughout the year; (4) intensity of musculoskeletal pain or discomfort. As the National Institute of Occupational Safety and Health (NIOSH) recommended [23], WMSD cases are required to satisfy all the following criteria: (1) Discomfort within the past year; (2) Discomfort began after employment in the current job; (3) No prior accident or sudden injury (affecting the focal area of discomfort); (4) Episodes of discomfort occur monthly or, if not every month, at least exceed a weeklong period of discomfort. The third part captures information on the exposure to ergonomics-related risk factors and labor organization factors. Exposure to ergonomics-related risk factors is assessed by the questions addressing awkward postures, manual material handling (MMH), excessive force exertion, contact stress (i.e., pressing a certain part of the body against hard or sharp edges, or using the hand as a hammer), repetitive movements, and vibration. Questions related to awkward posture included trunk flexion, trunk twisting, prolonged working posture (including standing, sitting, squatting, and kneeling), prolonged neck bending forward or twisting, prolonged bending wrist, prolonged bending knee, holding objects, or pinch grip. Manual material handling is assessed by the presence of carrying heavy loads >20 kg. Several questions are used to assess the presence of contact stress, repetitive movements, and vibration. Questions related to labor organization factors include overtime work, departmental staff shortage, and work–rest arrangements. Of note, in order to avoid missing data, the electronic questionnaire has been set to be submitted after all questions have been answered.

### 2.5. Statistical Analyses

The collected data were downloaded from our online database as a Microsoft Excel sheet and then exported to the Statistical Package for Social Sciences (SPSS) Version 22 (SPSS Inc., Chicago, IL, USA). Descriptive statistics were used to demonstrate the characteristics of study participants and the WMSDs distribution. Continuous variables were presented as the mean ± standard deviation (SD), and categorical data were presented as frequencies and percentages. Univariate and multivariate analyses were carried out to explore the potential risk factors associated with the development of WMSDs for each body region. The chi-square (χ^2^) test or Fisher’s exact test were used for univariate analysis to determine the association of individual, labor organization, and ergonomics-related risk factors with WMSDs. Any statistically significant factors in the univariate analysis were subsequently subjected to multivariate logistic regression analysis. Statistical significance was set at the 0.05 level. All *p*-values were two-sided.

### 2.6. Ethical Consideration

Ethical approval was obtained from the Medical Ethics Committee of the Guangzhou Twelfth People’s Hospital (NO. 2021024). Informed consent, included in the online questionnaire, was obtained from all the participants prior to the survey. The answers to the survey questions were anonymous, and the collected data were kept confidential.

## 3. Results

In total, 4181 people participated in the study; 2953 (70.63%) of them were male. The average age was 33.72 ± 7.52 years, with a range from 20 to 64. Half of the participants (51.42%) were aged 25 to 35. For BMI, this study adopted the criteria issued by the Ministry of Health of The People’s Republic of China in 2009. The BMI values of 460 participants (11%) were below 18.5 kg/m^2^; 2778 participants (66.44%) ranged from 18.5 to 23.9 (kg/m^2^); 791 participants (18.92%) ranged from 24.0 to 27.9 (kg/m^2^); the remaining 152 participants (3.64%) exceeded 28 kg/m^2^. Among all the surveyed subjects, 70.13% were engaged in furniture manufacturing for 1~2 years, 62.98% were married, 55.44% were in good physical health, and 46.26% had the habit of smoking. The proportion of those with a right or left dominant hand was 90.96% and 9.04%, respectively. In addition, the monthly income of most surveyed subjects (96.77%) exceeded RMB 3000, and only 9.43% finished their college or university education. The general characteristics of the participants are presented in Table 1.

### 3.1. Prevalence of the Work-Related Musculoskeletal Disorders

The incidences of WMSDs among furniture manufacturing workers, classified by body regions and gender, are shown in Table 2 and Figure 1. Overall, 31.57% (*n* = 1320) of the study participants reported that they had MSD symptoms in at least one body region within the previous 12 months. To be specific, the prevalence of WMSDs in the past 12 months among males was 31.36%, and that among females was 32.08%. Musculoskeletal disorder symptoms were most commonly reported in the neck (16.77%), followed by the shoulders (14.90%), ankles/feet (14.64%), hands/wrists (13.30%), upper back (11.48%), and lower back (10.95%). The least common sites reported were the elbows (9.81%), knees (10.00%), and hips/thighs (10.26%).

The gender difference of WMSDs in different body regions was tested, and the WMSDs in some body regions were found to be statistically significant, such as the shoulders (χ^2^ = 4.408, *p* = 0.036), hips/thighs (χ^2^ = 5.523, *p* = 0.019), knees (χ^2^ = 5.009, *p* = 0.025), and ankles/feet (χ^2^ = 4.776, *p* = 0.029) (Table 2).

### 3.2. The Risk Factors for WMSD Symptoms

Univariate and multivariate analysis were performed to identify the risk factors for WMSD symptoms. Appendix A present the results of the univariate analysis for each body region. The statistically significant variables identified from the univariate analysis were subsequently included in the multivariate logistic regression for further analysis. The adjusted risk estimates (odds ratio (OR) and 95% confidence interval (CI)) of the work-related factors for WMSD symptoms in the neck, trunk (upper back and lower back), upper extremities (shoulders, hands/wrists, elbows), and lower extremities (hips/thighs, knees, ankles/feet) were obtained by using the multivariate regression models with some significant demographic and social-economic confounders included. Table 3 shows the gender-pooled risk, while Table 4 presents the gender-specific risk estimates. 

#### 3.2.1. The Risk Factors for WMSD Symptoms in the Sites of the Neck and Trunk

The results of the multivariate analysis for WMSD symptoms in the neck, the upper back, and the lower back are shown in Table 3. The following factors were related to the development of WMSDs in the neck, the upper back, and the lower back, with *OR*s ranging from 1.32 to 3.63: *workers in moderate/poor/very poor physical health*, *working in an uncomfortable posture*, *exposure to cold, cool breeze or temperature changes when working*, *taking over another’s shift frequently*, *back bending forward when working* Two factors—*keeping the back in the same position for a long time* and *neck twisting for a long time while working*—significantly contributed to the development of WMSDs in the neck and upper back regions. The occurrence of WMSDs in the upper back and the lower back was significantly increased with *repeating the same movement on the trunk*, with *OR*s reported at 1.40 and 1.54, respectively. The neck symptom was statistically significantly associated with *doing the same job almost every day* (*OR* = 1.95, 95% CI: 1.30–2.92) and *department staff shortage* (*OR* = 1.27, 95% CI: 1.05–1.53). The factor *taking shift work* (*OR* = 1.28, 95% CI: 1.03–1.60) was significantly associated with MSD symptoms in the upper back, while *carrying heavy loads (more than 20 kg each time)* (*OR* = 1.32, 95% CI: 1.01–1.73), *trunk bending and twisting simultaneously* (*OR* = 1.45, 95% CI: 1.11–1.89), and *back bending over for a long time* (*OR* = 1.58, 95% CI: 1.26–1.99) were found to be the risk factors causing MSD symptoms in the lower back region. However, the multivariate regression model further revealed that *rotating jobs with colleagues* (*OR* = 0.71, 95% CI: 0.59–0.85) had a protective effect for the MSD symptoms in the neck, and *adequate rest time* significantly reduced the occurrence of MSD symptoms in the neck and the upper back regions (*OR* = 0.56, 95% CI: 0.46–0.68; *OR* = 0.65, 95% CI: 0.52–0.82).

#### 3.2.2. The Risk Factors for WMSD Symptoms in the Upper Extremities

The estimates of the risk factors for WMSD symptoms in the upper extremities (hands/wrists, elbows, and shoulders) are shown in Table 3 and Table 4. The following factors conferred a significant risk for WMSD symptoms in all the upper extremities in both genders: *workers in moderate/poor physical health* and *working in an uncomfortable posture*. The following factors significantly contributed to the development of WMSDs in the hands/wrists and elbows in both genders: *taking shift work*, *department staff shortage*, *bending wrists up and down frequently when working*, *bending wrist for a long time*, and *often placing wrists on the edge of hard and angular objects (e.g., a table edge)*. *Exposure to cold, cool breeze, or temperature changes when working* was associated with musculoskeletal disorders in the hands/wrists and elbows in both genders and in the shoulders in females. The results also revealed that *performing repetitive movements* was positively correlated with the occurrence of WMSDs in the hands/wrists in both genders (*OR* = 1.50, 95% CI: 1.16–1.93) and in the shoulders (*OR* = 1.39, 95% CI: 1.05–1.84) in males, while *adequate rest time* was linked to fewer reports of WMSD symptoms in the hands/wrists and shoulder regions in both genders, with *OR*s ranging from 0.609 to 0.717. *Taking over another’s shift frequently* contributed to elbow symptoms in both genders (*OR* = 1.71,95% CI: 1.30–2.25) and to shoulder symptoms in females (*OR* = 2.02, 95% CI: 1.26–3.22); *neck twisting for a long time while working* contributed to the genesis of MSDs in shoulders in both male (*OR* = 1.49, 95% CI: 1.15–1.94) and female (*OR* = 1.50, 95% CI: 1.02–2.20) workers. Moreover, our findings further indicated that *doing the same job almost every day* (*OR* = 1.81, 95% CI: 1.08–3.05), *trunk bending and twisting simultaneously* (*OR* = 1.36, 95% CI: 1.04–1.79), and *keeping trunk twisting for a long time* (*OR* = 1.49, 95% CI:1.14–1.95) were related to the shoulder symptoms in male workers, while *back bending forward when working* (*OR* = 1.59, 95% CI: 1.08–2.35) and *keeping the neck in the same position for a long time* (*OR* = 1.78, 95% CI: 1.17–2.70) were significantly associated with shoulder symptoms in female workers.

#### 3.2.3. The Risk Factors for WMSD Symptoms in the Lower Extremities

The results from the multivariate logistic regression analysis for the lower extremities (including hips/thighs, knees, and ankles/feet) are shown in Table 4. The following factors conferred significant risks for the development of WMSDs in the lower extremities, with *OR*s ranging from 1.35 to 5.34: *workers in moderate/poor physical health*, *working in an uncomfortable posture*, *exposure to cold, cool breeze, or temperature changes when working*, and *lower extremities often repeating the same movement*. In comparison, *department staff shortage* and *keeping knees bent for a long time when working* were statistically significantly linked to the lower extremities symptoms in male workers (*OR*s between 1.39 and 2.16), and *taking over another’s shift frequently* was found to be a labor organization factor for the lower extremities symptoms in female workers (*OR*s between 2.09 and 2.22). The symptoms in the hips/thighs and ankles/feet in males were statistically significantly associated with *prolonged sitting* and *taking shift work*, while the symptoms in the ankles/feet were statistically significantly associated with three additional work-related risk factors: *rotating jobs with colleagues* (*OR* = 0.78, 95% CI: 0.63–0.97), *performing repetitive movements* (*OR* = 1.34, 95% CI: 1.03–1.75), and *stretching or changing leg posture frequently when working* (*OR* = 1.59, 95% CI: 1.18–2.16). *Using vibration tools* and *performing repetitive movements* were also risk factors for the symptoms in knees in males, increasing the odds by 47.1% and 47.7%, respectively. As for female workers, *starting to work again after a break* (*OR* = 3.61, 95% CI: 1.09–11.94) and *type of work (other staff)* (*OR* = 0.18, 95% CI: 0.07–0.45) were the work-related factors for the knees and ankles/feet symptoms, respectively.

## 4. Discussions

The purposes of this study were to estimate the prevalence of work-related musculoskeletal disorders among manufacturing workers involved in furniture production and to investigate the risk factors for WMSDs in various body sites/regions. In the analysis, the overall prevalence of musculoskeletal disorders in any part of the body within the previous 12 months was 31.57%, consistent with the previous report [20] conducted in China. The highest prevalence of WMSDs among all the body regions was in the neck (16.77%), followed by the shoulders (14.90%), ankles/feet (14.64%), and hands/wrists (13.30%). Thetkathuek et al. [13] reported that, among 439 sampled Thailand furniture factory workers, the highest prevalence of MSD symptoms was in the shoulders (53.9%), followed by the hands/wrists (37.8%), upper back (37.5%), and lower back (35.9%). A survey conducted on 410 workers in small furniture manufacturing workshops located in Iran revealed that workers experience discomfort mostly in the knees (39.0%), followed by the lower back (35.6%), hands/wrists (29.5%), and shoulders (20.0%). Such discrepancies in the prevalence and region distribution of WMSDs were likely due to the difference in mechanization and working conditions [24].

With regard to individual factors, age was generally considered an important risk factor contributing to the development of WMSDs in many body regions. This positive association has been reported for samples of shipyard workers and construction workers [9,25,26]. Musculoskeletal physiology and structure may change with age, resulting in a decline in physical fitness and endurance to resist or withstand physical strains and stresses [27]. Similarly, career length was also reported to be an influential factor causing WMSDs in previous studies [13,28,29]. Unexpectedly, we did not observe a significant association between the risk of WMSDs in any parts of the body and increasing age or career length in the logistic analysis. 

Our findings suggested that female workers are more likely to develop WMSD symptoms in the shoulders area, which is in line with many published works in the literature [30,31,32]. Treaster et al. [33] suggested that the tasks of female workers are usually characterized by a high static loading of the shoulders area, requiring precision and the repetitive use of small muscles, which may account for the higher prevalence of WMSDs in the shoulders of female workers. Furthermore, compared to workers with a good health status, those with a relatively poor health status were two to three times more likely to develop WMSDs in all body regions, which is in agreement with the result of Alexopoulos et al. [25]. The latter found that bad/moderate general health increased the odds for WMSD symptoms in the lower back, hands/wrists, shoulders, and neck. Interestingly, our logistic analysis further revealed that workers engaged in the frontline production were more likely to suffer from WMSDs in the ankles/feet region. According to our on-site investigation, only about 5% of the frontline workers had a college degree or above, and they were mainly engaged in physical activities related to furniture production. The frontline workers had relatively high-intensity work tasks, which required a higher physical demand, thus resulting in more WMSDs in this area. Moreover, the work forms of other staff, such as technical management personnel, managerial and office workers, and auxiliary workers, were based on computer operation, paperwork, and other low-intensity physical activities. Managerial and office workers were mainly involved in sedentary work and rarely engaged in physical work tasks such as material handling, which was also the protection of lower extremities, to a certain extent.

As for labor organization factors, *starting to work again after a break*, *department staff shortages*, *doing the same job almost every day*, and *taking over another’s shift* were regarded as risk factors leading to the development of WMSDs, while *rotating jobs with colleagues* and *an adequate rest time* were considered as factors reducing the odds of WMSDs. As previous studies [34,35,36] illustrated, job rotation can eliminate the boredom and monotony related to simple repetitive tasks as well as increase motor variability, which may have a potential benefit for delaying or preventing acute fatigue and the development of chronic musculoskeletal disorders. *Department staff shortages* and *taking over another’s shift* often cause workers to take on more unscheduled work tasks and shorten the rest time. Lundberg et al. [37] found that a lack of adequate rest and recovery seems to be even more harmful for health than stress and physical overdraft during work. Prolonged working hours and insufficient rest time will render the body unable to fully recover after high-intensity work, and workers will be in a state of fatigue, which is more likely to induce the occurrence of WMSDs [38,39]. Likewise, doing the same job every day may cause cumulative fatigue in the same specific areas, thus leading to musculoskeletal damage easier. 

Numerous studies [40,41,42] have shown that taking shift work has a negative effect on workers’ mental health, often causing anxiety and depression, especially among female workers. Sleep deprivation related to this factor causes neurocognitive declines in function and performance, which further contribute to fatigue-related injuries and work errors [43,44]. Hence, health promotion programs or policies at the workplace are needed to optimize workers’ schedules and minimize the shift workers’ risk of poor mental health and injuries.

In agreement with our research findings for the ergonomics-related risk factors, several studies [13,16,38] reported that carrying heavy loads, using vibration tools, and performing repetitive movements were significant risk factors for MSD symptoms among manufacturing workers. However, it was an unanticipated finding that using vibration tools increased the odds of WMSDs in the knees region rather than in the hands/wrists or lower back regions, as reported in previous studies [45,46]. According to our on-site observations, when handling material, the forklift drivers need to operate the machine in a standing position rather than the traditional seated position, which may bring a larger load on their lower limbs. This finding may lead to an assumption that the biomechanical load on the knees from standing, combined with whole-body vibrations during operations, may exacerbate musculoskeletal damage to the areas mentioned above [46,47]; however, this needs further field tests with suitable methods such as electromyogram (EMG) signal analysis. It should be noted that cold temperatures combined with vibration and force exertion may increase the risk of musculoskeletal disorders.

*Back bending forward when working*, *trunk bending and twisting simultaneously*, *repeating the same movements on the trunk*, *keeping the back in the same position for a long time, bending over for a long time*, and *neck twisting for a long time while working*, which are common non-neutral postures in furniture production, were found to be risk factors for the occurrence of WMSDs in the neck, upper back, and lower back. According to the anatomy of the human body, cervical, thoracic, and lumbar spine segments start from the neck and go down to the tail bone, and the prolonged deviation of the spine from its normal curvature causes the body limb to develop excessive muscle strain and bone stresses, which may contribute to the development of WMSDs in the neck and trunk areas [28]. The non-neutral postures in the neck and back also contributed to the development of WMSDs in the shoulders. This finding is consistent with other previous studies, supporting the idea that non-neutral postures involving bending and twisting increase the risk of WMSD symptoms in the shoulders [48,49,50]. The non-neutral position of the spine may cause the destabilization of the shoulder joint and the increased activation of the shoulder muscles [51].

*Bending wrists up and down frequently when working* and *often placing wrists on the edge of hard and angular objects (e.g., a table edge)* are commonly observed in computer operation and manual works. For one thing, the frequent bending of the wrists may lead to repeated changes in the carpal tunnel pressure, increasing the risk of carpal tunnel syndrome (CTS) [52]. Further, repetitive movements of the wrists and forearm/elbow muscles may lead to the development of musculoskeletal damage to the wrists and elbows. *Often placing wrists on the edge of hard and angular objects* would cause a contact stress to the hands/wrists and elbows, while bending wrists for a long time can be considered as a continuous muscle contraction process, which will cause muscle tension in the elbows and wrists, resulting in muscle ischemia and fatigue in corresponding parts. Additionally, da Costa et al. [14] also identified repetition and awkward posture as significant factors causing WMSDs in the lower extremities, which is partly consistent with our study’s finding that *stretching or changing leg posture frequently when working, keeping knees bent for a long time when working*, and *often repeating the same movement* increase the odds of MSD symptoms.

## 5. Limitations

There are several limitations in this survey that are worth noting. First and foremost, since this was a cross-sectional study, any causal inferences should be drawn with caution. Moreover, the data collection was based on workers’ self-reports and relies heavily on their subjective feelings rather than on objective measurements (e.g., force measurements, biomechanical modeling, and/or determination by using the muscle injury-related biomarkers), which may lead to some reporting bias. Recall bias may have occurred when participants were asked to report on the frequency and location of WMSDs in the previous 12 months, which could be another limitation. This study was conducted among workers in one large-scale furniture manufacturer; therefore, extensive generalization cannot be made. Finally, most of the findings from this large-scale survey were consistent with those of several previous studies, merely demonstrating the validity of many existing studies. Nevertheless, in order to translate these findings into practice, cumulative evidence from various studies and systematic meta-analyses of the existing studies are highly demanded.

## 6. Conclusions

This study has quantitatively evaluated the WMSD prevalence and its risk factors among furniture manufacturing workers. In general, the evidence from the results indicates that the neck, shoulders, ankles/feet, and hands/wrists are primarily the most affected body regions. Moreover, multiple risk factors including individual, labor organization, and ergonomics-related factors were found to significantly contribute to the development of WMSDs in different body regions. Considering the magnitude of the issue, there is an urgent need for educational intervention for the employees of the furniture manufacturing industries in order to provide them with adequate ergonomic knowledge. For example, it is advisable to instruct them to work in a neutral position and to avoid maintaining the same posture for long periods of time. In addition, based on the results of the study, several cost-effective solutions can be recommended for policymakers to alleviate the health burden caused by WMSDs. These include developing reasonable work–rest cycles and job rotation schedules, increasing the labor quota and break time, adding adjustable seats for forklifts, controlling the workplace environment, etc.

## Figures and Tables

**Figure 1 ijerph-19-14435-f001:**
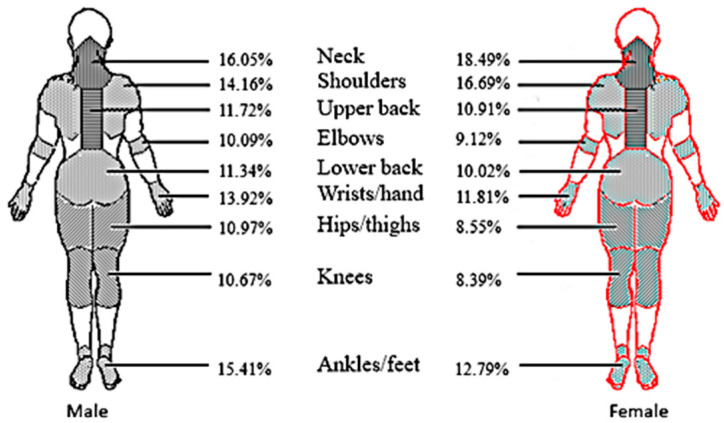
Gender-specific distribution of different musculoskeletal regions impaired among furniture manufacturing workers. Male workers are shown in black and female workers are shown in red.

**Table 1 ijerph-19-14435-t001:** Demographic and work-related characteristics of the participants (N = 4181).

Items	Total		Male		Female	
	*n*	%	*n*	%	*n*	%
Age (years)						
<25	454	10.86	365	8.73	89	2.13
25~	2150	51.42	1576	37.69	574	13.73
35~	1261	30.16	817	19.54	444	10.62
45~	316	7.56	195	4.66	121	2.89
BMI (body mass index, kg/m^2^)						
<18.5	460	11.00	265	6.34	195	4.66
18.5~	2778	66.44	1932	46.21	846	20.23
24.0~	791	18.92	638	15.26	153	3.66
28.0~	152	3.64	118	2.82	34	0.81
Career length (years)						
1~2	2932	70.13	2045	48.91	887	21.22
3~5	898	21.48	657	15.71	241	5.76
6~10	248	5.93	180	4.31	68	1.63
10~	103	2.46	71	1.70	32	0.77
Educational level						
Junior high school and below	2569	61.44	1685	40.30	884	21.14
High school or technical secondary school	1218	29.13	985	23.56	233	5.57
College or university	382	9.14	276	6.60	106	2.54
Postgraduate degree or above	12	0.29	7	0.17	5	0.12
Marital status						
Never married	1422	34.01	1247	29.83	175	4.19
Married	2633	62.98	1617	38.67	1016	24.30
Else (divorced or widowed)	126	3.01	89	2.13	37	0.88
Monthly income						
RMB ≤ 1000	27	0.65	23	0.55	4	0.10
RMB 1001–3000	108	2.58	56	1.34	52	1.24
RMB 3001–5000	1802	43.10	1085	25.95	717	17.15
RMB > 5000	2244	53.67	1789	42.79	455	10.88
Physical exercise						
Never	1503	35.95	990	23.68	513	12.27
Sometimes	2181	52.16	1574	37.65	607	14.52
2~3 times a month	160	3.83	123	2.94	37	0.88
1~2 times a week	216	5.17	165	3.95	51	1.22
>3 times a week	121	2.89	101	2.42	20	0.48
Smoking habits						
Non-smokers	2247	53.74	1034	24.73	1213	29.01
Smokers	1934	46.26	1919	45.90	15	0.36
Physical health status						
Good	2318	55.44	1636	39.13	682	16.31
Moderate	1671	39.97	1180	28.22	491	11.74
Poor	151	3.61	112	2.68	39	0.93
Very poor	41	0.98	25	0.60	16	0.38
Dominant hand						
Right	3803	90.96	2648	89.67	1155	94.06
Left	378	9.04	305	10.33	73	5.94
Type of work						
Frontline workers	3655	87.42	2605	88.21	1050	85.50
Other staff *	526	12.58	348	11.78	178	14.50

* including technical management personnel and auxiliary workers.

**Table 2 ijerph-19-14435-t002:** Prevalence of WMSDs among furniture manufacturing workers, classified by body regions and gender.

Body Site	All Participants		Male		Female		χ^2^	*p*-Value ^a^
	*n*	%	*n*	%	*n*	%		
Overall *	1320	31.57	926	31.36	394	32.08	0.21	0.64
Neck	701	16.77	474	16.05	227	18.49	3.68	**0.05**
Shoulders	623	14.90	418	14.16	205	16.69	4.40	**0.03**
Ankles/Feet	612	14.64	455	15.41	157	12.79	4.77	**0.02**
Hands/Wrists	556	13.30	411	13.92	145	11.81	3.35	0.06
Upper back	480	11.48	346	11.72	134	10.91	0.55	0.45
Lower back	458	10.95	335	11.34	123	10.02	1.56	0.21
Hips/Thighs	429	10.26	324	10.97	105	8.55	5.52	**0.01**
Knees	418	10.00	315	10.67	103	8.39	5.00	**0.02**
Elbows	410	9.81	298	10.09	112	9.12	0.92	0.33

^a^: *p* value for comparing two genders; statistically significant results are written in bold. * suffering from a problem(s) in at least one body region.

**Table 3 ijerph-19-14435-t003:** The adjusted odds ratios (95% CI) of the risk factors for WMSDs in the neck, trunk, hands/wrists, and elbows.

Factors	Odds Ratios (95% Confidence Interval)				
	Neck	Upper Back	Lower Back	Hands/Wrists	Elbows
Physical health status					
Good	1.00	1.00	1.00	1.00	1.00
Moderate	1.80 (1.50, 2.17)	1.63 (1.32, 2.02)	2.10 (1.69, 2.62)	1.42 (1.16, 1.73)	1.58 (1.26, 1.98)
Poor	3.34 (2.29, 4.86)	2.75 (1.80, 4.20)	3.63 (2.38, 5.53)	1.97 (1.28, 3.03)	1.97 (1.22, 3.18)
Very poor	3.12 (1.52, 6.38)	2.52 (1.16, 5.48)	2.97 (1.38, 6.41)	1.99 (0.91, 4.34)	2.95 (1.36, 6.43)
Carrying heavy loads (more than 20 kg each time)			1.32 (1.01, 1.73)		
Working in an uncomfortable posture	1.95 (1.61, 2.35)	1.99 (1.58, 2.49)	2.20 (1.73, 2.80)	1.54 (1.25, 1.89)	1.89 (1.48, 2.40)
Doing the same job almost every day	1.95 (1.30, 2.92)				
Rotating jobs with colleagues	0.71 (0.59, 0.85)				
Performing repetitive movements				1.50 (1.16, 1.93)	
Exposure to cold, cool breeze, or temperature changes when working	1.48 (1.21, 1.80)	1.38 (1.10, 1.72)	1.35 (1.08, 1.69)	1.41 (1.14, 1.73)	1.54 (1.22, 1.94)
Taking shift work		1.28 (1.03, 1.60)		1.51 (1.22, 1.87)	1.50 (1.18, 1.91)
Adequate rest time	0.56 (0.46, 0.68)	0.65 (0.52, 0.82)		0.71 (0.58, 0.88)	
Department staff shortage	1.27 (1.05, 1.53)			1.35 (1.11, 1.65)	1.31 (1.03, 1.65)
Taking over another’s shift frequently	1.37 (1.07, 1.75)	1.76 (1.37, 2.27)	1.45 (1.11, 1.89)		1.71 (1.30, 2.25)
Back bending forward when working	1.49 (1.16, 1.90)	1.32 (1.04, 1.66)	1.33 (1.04, 1.71)	-	-
Trunk bending and twisting simultaneously			1.45 (1.11, 1.89)	-	-
Repeating the same movement on the trunk		1.40 (1.06, 1.83)	1.54 (1.14, 2.06)	-	-
Keeping the back in the same position for a long time	1.62 (1.29, 2.03)	1.44 (1.13, 1.84)		-	-
Back bending over for a long time			1.58 (1.26, 1.99)	-	-
Neck twisting for a long time while working	1.45 (1.16, 1.81)	1.89 (1.51, 2.37)		-	-
Bending wrists up and down frequently when working	-	-	-	1.73 (1.25, 2.41)	2.07 (1.42, 3.03)
Bending wrist for a long time	-	-	-	2.01 (1.59, 2.53)	1.86 (1.42, 2.42)
Often placing wrists on the edge of hard and angular objects (e.g., a table edge)	-	-	-	1.91 (1.51, 2.42)	1.62 (1.24, 2.11)

-: not applicable for the body part.

**Table 4 ijerph-19-14435-t004:** The adjusted odds ratios (95% CI) of the risk factors for WMSDs in the shoulders and lower extremities.

Factors	Shoulders		Hips/Thighs	
	Male	Female	Male	Female
Physical health status				
Good	1.00	1.00	1.00	
Moderate	2.18 (1.73, 2.75)	1.62 (1.16, 2.27)	2.22 (1.70, 2.90)	
Poor	3.66 (2.31, 5.77)	2.21 (1.03, 4.72)	4.09 (2.50, 6.70)	
Very poor	2.24 (0.84, 6.00)	2.45 (0.78, 7.66)	5.34 (2.16, 13.19)	
Prolonged sitting			0.59 (0.44, 0.80)	
Working in an uncomfortable posture	1.90 (1.49, 2.43)	1.79 (1.28, 2.51)	1.81 (1.37, 2.40)	2.92 (1.90, 4.48)
Doing the same job almost every day	1.81 (1.08, 3.05)			
Performing repetitive movements	1.39 (1.05, 1.84)			
Exposure to cold, cool breeze, or temperature changes when working		1.55 (1.06, 2.27)	1.58 (1.22, 2.06)	
Taking shift work			1.37 (1.04, 1.80)	
Adequate rest time	0.71 (0.56, 0.91)	0.60 (0.43, 0.85)	0.75 (0.57, 0.99)	0.59 (0.38, 0.92)
Department staff shortage			1.55 (1.20, 2.00)	
Taking over another’s shift frequently		2.02 (1.26, 3.22)		2.22 (1.28, 3.87)
Back bending forward when working		1.59 (1.08, 2.35)	-	-
Trunk bending and twisting simultaneously	1.36 (1.04, 1.79)		-	-
Keeping trunk twisting for a long time	1.49 (1.14, 1.95)		-	-
Keeping the neck in the same position for a long time		1.78 (1.17, 2.70)	-	-
Neck twisting for a long time while working	1.49 (1.15, 1.94)	1.50 (1.02, 2.20)	-	-
Keeping knees bent for a long time when working	-	-	2.16 (1.63, 2.86)	
Lower extremities often repeating the same movement	-	-	1.82 (1.36, 2.43)	
**Factors**	**Knees**	**Ankles/Feet**		
	**Male**	**Female**	**Male**	**Female**
Physical health status				
Good	1.00	1.00	1.00	1.00
Moderate	1.99 (1.52, 2.60)	1.78 (1.13, 2.80)	1.89 (1.51, 2.37)	1.57 (1.08, 2.27)
Poor	3.63 (2.21, 5.97)	4.83 (2.01, 11.60)	2.95 (1.87, 4.66)	3.40 (1.54, 7.51)
Very poor	3.56 (1.35, 9.35)	3.86 (0.96, 15.51)	3.57 (1.46, 8.74)	0.94 (0.19, 4.55)
Type of work				
Frontline workers				1.00
Other staff *				0.18 (0.07, 0.45)
Prolonged sitting			0.74 (0.57, 0.96)	
Using vibration tools	1.47 (1.13, 1.91)			
Working in an uncomfortable posture	1.49 (1.12, 2.00)	1.90 (1.21, 2.98)	1.89 (1.49, 2.40)	1.99 (1.38, 2.87)
Rotating jobs with colleagues			0.78 (0.63, 0.97)	
Performing repetitive movements	1.47 (1.07, 2.02)		1.34 (1.03, 1.75)	
Exposure to cold, cool breeze or temperature changes when working	1.49 (1.14, 1.95)	1.64 (1.01, 2.67)	1.35 (1.07, 1.71)	
Taking shift work			1.35 (1.07, 1.70)	
Often working overtime				
Adequate rest time	0.64 (0.48, 0.86)	0.56 (0.35, 0.90)	0.69 (0.55, 0.88)	0.55 (0.37, 0.81)
Starting working again after a break		3.61 (1.09, 11.94)		
Department staff shortage	1.62 (1.26, 2.10)		1.39 (1.11, 1.73)	
Taking over another’s shift frequently		1.87 (1.03, 3.38)		2.09 (1.25, 3.50)
Stretching or changing leg posture frequently when working			1.59 (1.18, 2.16)	
Keeping knees bent for a long time when working	1.75 (1.32, 2.32)		1.42 (1.10, 1.83)	
Lower extremities often repeating the same movement	1.69 (1.25, 2.29)	1.68 (1.08, 2.61)	1.55 (1.21, 1.99)	1.74 (1.21, 2.50)

* other staff employees included technical management personnel and auxiliary worker; -: not applicable for the body part.

## Data Availability

The data are available from the authors on request.

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
