# Peer review of "Prevalence of Musculoskeletal Disorders and Their Associated Risk Factors among Furniture Manufacturing Workers in Guangdong, China: A Cross-Sectional Study"

_ijerph, 2022, doi:10.3390/ijerph192114435_

Round 1

Reviewer 1 Report

This document is an excellent study, well done. The topic is of importance given the large number of musculoskeletal problems that frequently occur in furniture manufacturers.

  If carried out with academic rigor, this article has the potential to be of value to practitioners and policymakers on the prevalence, risk factors, and prevention of musculoskeletal disorders in furniture manufacturers.

Also, in my opinion, the topic and premise of the study would fit well in the journal to which it was submitted. The authors are to be congratulated for conducting this study; however, there are some issues with the manuscript that require attention prior to publication. These will be discussed below in relation to sections of the manuscript:

- Title: The title is correct as it reflects correctly the objective of the work.

- Summary: Correct

- I recommend putting the keywords in lowercase letters, all the ones that are not proper names.

- Introduction: The research question itself is sound, however overall the introduction would benefit from adding more detail on what the problem is, how much it affects people and the economic implications of this would add depth to the question. introduction. Describe the hypothesis in this section.

- Materials and Methods: The methodology used is correct and referenced by adequate bibliography. The inclusion and exclusion criteria appear robust. It is reported that the participants gave their appropriate consent and that the protocol was approved by an ethics committee. But I miss the sample calculation, please, you must add them.. 

- Results: The data is presented adequately and correctly. The way of exposing them divided and according to regions of the body clarifies them and gives them a lot of order for understanding.

- Discussion: In general, the discussion of the study results is well balanced and identifies both strengths and possible weaknesses and compares them with other studies.

Reviewer 2 Report

MDPI_ ijerph-1913894-peer-review-v1   Reviewer 1 (Round 1)  Comments

 It is evident that the working environment and health of the subjects of this manuscript are easily overlooked. The purpose of this research goals and ideas are relatively clear, and the authors have also chosen an appropriate method of conducting experiments and analyzing data. However, before it could be published, it may need to be refined in the following areas:

1.     Section 2.2

Subjects were randomly selected from the 11 factories affiliated with the 4 large--scale furniture manufacturer in Guangdong province.

Perhaps due to objective conditions, only one factory was chosen from 11 candidates. Is the situation the same in all 11 factories? If so, then it would be reasonable to choose one at random. If not, the author needs further explanation.

2.     Section 2.4

The author mentions: After some modifications to adapt Chinese characteristics…

So, do these changes have any supporting literature? Statistical significance is one of the most basic requirements, but in addition, more literature must be cited to demonstrate reliability and validity.

3.     Chapter 4

It is evident that the researchers have considerable experience in analyzing questionnaire data. In the fourth chapter, although the authors make a few new discoveries, they are merely demonstrating the validity of many existing studies. The contribution of this article is greatly diminished as a result.

Possibly, this can be explained by the limitations discussed in chapter 5. This also leaves room for improvement in chapter 6. For example, it is important that the authors explain what preventive measures are available based on their findings.

4.     Minor issues

There is a mistake in the numbering of the third-level heading in chapter 3. This should be changed to: 3.2.3 The risk factors for WMSDs symptoms in the lower extremities.

Formatting must be completed according to the journal’s format template. In addition, some references have incomplete information (e.g., No. 3).

The format of some references is incorrect (journal names are not abbreviated).

Round 2

Reviewer 2 Report

I am glad to the authors explain or correct the issues identified during the previous round of review.  Ihas been noted previously that the authors’ re-proof of existing findings will be the foundation for future research. I  am looking forward to the authors' follow-up research.

However, the format of the paper still needs to be improved since it is not standardized.